# Deployment of Solar Energy at the Expense of Conservation Sensitive Areas Precludes Its Classification as an Environmentally Sustainable Activity

**Francisco Valera [1],*, Luis Bolonio [1], Abel La Calle [2] and Eulalia Moreno [1]**

[1] Departamento de Ecología Funcional y Evolutiva, Estación Experimental de Zonas Áridas (EEZA-CSIC), Ctra. de Sacramento s/n, La Cañada de San Urbano, E-04120 Almería, Spain

[2] Departamento de Derecho, Universidad de Almería, Ctra. de Sacramento s/n, La Cañada de San Urbano, E-04120 Almería, Spain

* Correspondence: pvalera@eeza.csic.es

**Abstract:** Solar energy (SE) is essential for the decarbonization of our economy and for energetic transition. Solar energy can be a sustainable economic activity, as long as a balance is struck between the benefits it brings to climate change mitigation and the damage it can cause to biodiversity and ecosystems. Here, we study this balance in an area with high biodiversity under pressure for installation of numerous photovoltaic plants (PPs). Our results show that developers give priority to the cheapest land close to connection points, while other values (e.g., environmental, landscape) are secondary. The regulatory process carried out by the Administration does not ensure the preservation of natural values, as several PPs with a high impact on important conservation areas have been approved. Experts' allegations provide quality information to the Administration to evaluate and demand changes to the projects presented. Such demands show that companies are willing to relocate plants to land occupied by olive groves. In this way, greater efficiency is achieved in land occupation, as well as shorter evacuation lines, water savings and a smaller environmental impact. Prior strategic territorial planning could have avoided the impact of PPs already built, made the deployment of new PPs compatible with biodiversity conservation, and contributed to improving the management of key resources, such as subway aquifers. The proposed regulatory changes to the environmental assessment procedure (exclusion of renewables and public participation from the procedure) are detrimental, as they will make SE unable to meet the requirements of the Taxonomy Regulation.

**Keywords:** sustainability; solar energy; land-use; taxonomy regulation; territorial planning; biodiversity





## 1. Introduction

Climate change (CC) is the biggest challenge of our times, and the need to reduce greenhouse gas emissions is a world-wide priority. A global energy transition is urgently needed to meet the objectives of limiting the average global surface temperature increase to below 2 °C. Energy efficiency and renewable energy technologies (namely solar photovoltaics and wind power) are the core elements of that transition [1]. Photovoltaic solar energy (PSE) is recognized as one main way to produce energy with a lower carbon footprint, due to its low greenhouse gas emissions [2].

The deployment of PSE in the European Union (EU) must be undertaken within the framework of Regulation (EU) 2020/852 (known as the Taxonomy Regulation, Text S1), a key piece in the EU's Sustainable Finance Plan, as it defines harmonized criteria to qualify an economic activity as environmentally sustainable. The Regulation establishes that an economic activity will be considered environmentally sustainable when it contributes substantially to one or more of the environmental objectives set out in Article 9, including CC mitigation and adaptation to CC.

PSE meets these requirements and therefore qualifies for being a sustainable economic activity. However, the United Nations Economic Commission for Europe also indicates that it is the technology with the greatest impact on land occupation (after concentrated solar power), and that it can reach 40 m$^2$ per megawatts and hour (MWh) produced [2]. In Spain, the Ministry for the Ecological Transition and the Demographic Challenge (MITECO) warns that photovoltaic plants (PPs), due to their high degree of occupation and transformation of territory, can have a direct impact on: (i) biodiversity, as it implies the loss and destruction of flora and fauna habitats and the fragmentation of populations, and (ii) the landscape, by introducing a large number of anthropic elements in a matrix with a high degree of naturalness. This indicates that implementing this technology, without carrying out an adequate and prior spatial planning, may have adverse effects (Regulation (EU) 2020/852, Article 3b) that question its classification as an environmentally sustainable activity. In fact, the Taxonomy Regulation establishes that, for an economic activity to be considered environmentally sustainable, it must not cause any significant detriment to any of the environmental objectives established in Article 9, among which are protection and recovery of biodiversity and ecosystems.

The implementation of PSE in Spain is proceeding at a frenetic pace. In 2019, it was the European country where the largest solar photovoltaic generation capacity was installed, with 4.7 gigawatts (GW) installed out of the continental total of 13.4 GW, and in 2020 was the second leading European country for newly installed photovoltaic capacity, with 2.7 GW [3]. As of 31 October 2022, there are 14.3 GW of photovoltaic solar capacity in service [4]. Spain's goal for 2030 is to multiply its current deployment by 4 times, which implies a target of 3000 MW/year, to achieve 39 GW in service by 2030, according to the objectives of the National Energy and Climate Plan 2030 [5]. These objectives, together with the lower costs associated with efficient power generation from the Sun, have exponentially boosted investment interest at the national and international levels. As of 31 October 2022, throughout Spain there are 101.7 GW with access permits and 22.8 GW with access permit applications in process [4]. In total, there will be 138.8 GW, which is 3.4 times more than what is forecast for 2030 by the PNIEC. Andalucía is the Spanish region with the highest concentration of PP projects, with 3.1 GW in operation, 23.2 GW with access permits and 6.1 GW with access requests [4]. This comes to 32.4 GW (if all the projects are completed), which is very close to the 39 GW that the PNIEC indicates as a target to be reached in 2030 for the whole of Spain.

This whirlwind of projects gives rise to numerous problems. These range from practical aspects, such as the lack of government resources to adequately supervise the environmental impact studies commissioned by energy companies, to more fundamental aspects, such as the need for national energy planning to reduce the environmental impact of the large-scale deployment of PPs. One of the most important impacts, caused by the massive and disorderly occupation of the territory by PPs, is the effect these infrastructures have on diversity [6]. The lands with the best aptitudes for the installation of PPs (plain areas, low economic value, and proximity to the transport network) usually coincide with well-preserved steppe habitats important for wildlife. One clear example of such conflict is evidenced by steppe birds, many of which are threatened and/or in regression in Spain [7]. These species are adapted to live in high visibility conditions and shy away from vertical structures such as photovoltaic panels. There is clear evidence of the negative impact of PPs and their evacuation lines on these and other species [8–10]. Importantly, Spanish responsibility for the protection of these species is paramount, given that Spain has long since been the main European refuge for most of these birds [11,12]. According to Regulation 2020/852, an unplanned deployment of PSE cannot be considered environmentally sustainable if it causes significant harm to the conservation (protection and restoration) of biodiversity and ecosystems (Article 17, point f. ii), and in particular to species and sites of high interest to the EU.

While the occupation of large tracts of land by PPs is a global phenomenon [8], this process mainly affects some regions. Because of its high level of solar radiation, Iberian

southeast drylands are a particularly target. This is a warm semi-arid region (mostly including the provinces of Almería and Murcia), where the proposals submitted to the Spanish administrations for the construction of PPs have soared to at least 2.3 GW in the province of Almeria alone. This is the most arid region of continental Europe and is among the richest European regions in plant species (ca. 3000 sp., including mountain ranges) with abundant local or Iberian–North African endemic species, hosting a diverse and singular fauna [13–15]. Specifically, Campo de Tabernas (Almería province) is a paradigmatic territory in the context of the ecological transition, as it concentrates projects that add up to 1.7 GW of PSE. On the other hand, Campo de Tabernas is an excellent example of the arid ecosystems of SE Spain (probably the best preserved of the entire European continent), and has marked environmental, ecological and landscape values [16–18].

This paper analyzes the deployment of renewable energies, namely PPs, in this region, as well as their harmonization with the environmental objectives of the ecological transition set out in Regulation (EU) 2020/852 of the European Parliament. Specifically, we pursue the following aims: (i) to describe the main characteristics of solar energy (SE) deployment in a particularly attractive area for such a purpose; (ii) to evaluate its impact on protected areas and areas of special interest for biodiversity conservation. Additionally, we evaluate the importance of the involvement of experts and academics in the public consultation phase of the process regulating the arrangement of SE. We expect that the effectiveness of such a process, in terms of protecting endangered species and their habitats, will be improved by the incorporation of scientific knowledge. Finally, we make specific proposals for a deployment of PPs which is compatible with biodiversity conservation and estimate the pros and cons (in terms of water savings, important biodiversity areas affected by PPS) in this ecologically fragile and valuable environment.

## 2. Methods

### 2.1. Study Area

The study area lies in Campo de Tabernas (Almería), an excellent example of the semi-arid ecosystems of southeastern Spain. It is a landscape organized around the Rambla de Tabernas and strongly conditioned by the arid climate—its geomorphology and strongly conditioned by the arid climate and by its geomorphology and vegetation. It occupies several municipalities, the main ones being Turrillas (37°02′ N, 2°16′ O, 39 km$^2$), Lucainena de las Torres (37°02′ N 2°11′ O, 123.1 km$^2$), Sorbas (37°05′ N 2°07′ O, 249 km$^2$) and Tabernas (37°03′ N 2°23′ O, 281 km$^2$). These municipalities occupy ca. 69,000 ha.

The climate is semi-arid Mediterranean, with a strong water deficit in summer. The mean annual rainfall is approximately 230 mm, with high interannual and intra-annual variability [19]. The average annual temperature is 18 °C, with mild interannual oscillations and significant intra-annual fluctuations [20]. The province of Almeria is among the first in Spain in terms of solar irradiation (an average of 5.4 kWh/m$^2$ of direct radiation per day in Tabernas [21]).

The area consists of badlands with olive and almond groves, and cereal crops, interspersed among dry streambeds (ramblas) and steppe habitats with natural vegetation. Water scarcity has discouraged exploitation by humans to some extent, so that until the 1950s only about 200 ha of olive groves were irrigated in this area [22]. However, irrigation has expanded to 4400 ha in the last two decades. Moreover, a process of intensification has given way to super-intensive irrigation, which involves going from 210 to 1550 trees/ha, which in a few years has occupied more than 1500 ha [22]. Yet, there are still well-preserved spots with xerophytic plant communities [23]. The area (potentially) covered by Habitats of Community Interest (HCI) is more than 70% of the region and has 15 potential HCIs (of the total of 57 potential HCIs registered in Andalusia) [18].

Campo de Tabernas borders to the south with the Special Area of Conservation (SAC, ZEC in Spanish) "Ramblas de Gérgal, Tabernas and south of Sierra Alhamilla" (ES6110006) and with the Natural Site, the Special Protection Area for Birds (SPA, ZEPA in Spanish) and SAC "Sierra Alhamilla" (ES0000045), to the west with the Natural Site, SPA and SAC

"Desierto de Tabernas" (ES0000047) and to the east with the SAC "Sierra de Cabrera-Bédar" (ES6110005). These have the status of protected natural areas of the Natura 2000 network. Campo de Tabernas constitutes a natural ecological corridor connecting these areas. It is a key zone for preserving both unique and singular taxa from local extinction, as well as the ecological flows between their populations (see below Master Plan for the Improvement of Ecological Connectivity in Andalusia [18]).

The Plataforma Solar de Almería, a research center for concentrating solar power technologies under the auspices of the Centro de Investigaciones Energéticas, Medioambientales y Tecnológicas (CIEMAT), was built in 1977–1981 and occupies 99 ha.

The electric substation Tabernas 220 kv/400 kv is located in the study area; this is owned by Red Eléctrica Española (REE), the company in charge of high-voltage electricity transmission in Spain. Further away (ca. 48 km), there is a second electrical substation, La Ribina 400 kv, also owned by REE.

### 2.2. Photovoltaic Projects in the Study Area

Electricity production through PPs in Spain requires authorizations for transport and for construction and operation (art. 33 Act 24/2013, Text S1). If power exceeds 50 MW, authorizations are granted by the General State Administration, otherwise they are granted by the regional Authority (art. 3 Act 24/2013). In the study area, it is the latter (Autonomous Community of Andalusia) that authorizes most of the PPs. The Andalusian procedure of environmental prevention for PPs is the "Unified Environmental Authorization" (art. 19.3 Act 2/2007, Text S1), whose main phases (Decree 356/2010, Text S1) are: (i) application, accompanied by technical project, Environmental Impact Study (EIS) and urban compatibility; (ii) the Environmental Authority (EA) verifies the application and carries out a public consultation with publication in official journals and in the electronic headquarters; (iii) the EA examines the documentation and the result of the public consultation, and issues the Environmental Opinion that consults with the energy authority and interested parties; (iv) the EA examines the result of the consultations and issues a Binding Report proposal that only consults with the energy authority, after which it issues the Binding Report (favourable/unfavourable); (v) the EA adopts the final decision of granting or denying of the Unified Environmental Authorization. The procedure generally lasts more than one year. The procedure followed by the central authority is somewhat simpler (it unifies steps iii and iv).

During 2019–2022, we periodically reviewed the relevant official gazettes of the Spanish state, of the autonomous community and of the province (BOE, BOJA and BOP in Spanish, respectively) in search of applications for authorization for PPs in Almería province, with special attention to those to be developed in Campo de Tabernas. The EIS of these projects were obtained and analyzed in search of their main characteristics. Information on other projects in Campo de Tabernas prior to our study (mostly applications presented in 2017 and 2018) was obtained from various associations that presented allegations, as well as from official sources (BOJA, BOP).

This reviewing process provided information about: (i) characteristics of PPs proposed by enterprises (location, extension, and distribution of solar modules; type and value of the land chosen for the facility; length of the evacuation line and distance to the electric nodes (i.e., connection points)); (ii) problems associated with the proposals of PPs by the promoters, as evidenced by the allegations received (fragmentation, divergences with urban planning regulations, impact on wildlife, conflicts with the local population and with social organizations, etc.); and (iii) the response of the Administration and the enterprises to such allegations through the analysis of the Binding Reports.

As a result, information was obtained on 27 applications, involving 31 PPs planned or built in the study area. Additionally, we obtained information from the enterprises and their EISs about 4 PPs with right of access to the high-voltage network in Tabernas. In summary, the information provided here refers to 35 PPs in the study area (Table S1).

*2.3. Assessment of the Land Characteristics Chosen by the Developers to Install the PPs*

To analyze the preferences of the developers in terms of land type, we used the land use map of Spain corresponding to the European CORINE Land Cover (CLC) project, with a nomenclature of 44 classes (2018 version, [24]). The land uses in the study area, according to the CLC codes, are: 211—rainfed arable lands; 222—fruit trees, 223—olive groves; 231—meadows; 242—mosaic of crops; 243—predominantly agricultural land with important areas of natural vegetation; 321—natural grassland; 323—sclerophyllous vegetation; 333—areas with sparse vegetation.

Given the limitations of the information provided by CLC [25], these data were validated by direct observation in the field, so that the original categories were grouped by their similar characteristics and economic value into four groups. Olive groves (223) were divided into (1) super-intensive (ca. 1550 trees/ha) and (2) intensive (210 trees/ha) olive groves [22]. (3) Almond trees included both fruit trees (222) and mosaic of crops (242). (4) Extensive rainfed cereal crops with fallow land, grasslands and esparto grass (*Stipa tenaccisima* formations) included rainfed arable land (211), natural grasslands (321), meadows (231), sclerophyllous vegetation (323), areas with sparse vegetation (333) and agricultural land with natural vegetation (243).

To estimate the value of the soils chosen by developers to install PPs, the survey of land prices in Andalusia was consulted [26]. In the province of Almeria, this survey reports the maximum, minimum and most frequent prices for the following types of crops: irrigated winter vegetables, irrigated nut trees, rainfed nut trees, irrigated open-air vegetables, irrigated citrus fruits, irrigated olive trees for olive oil, and other crops used on pasture land. We have taken into account the most frequent price, assigning to categories 1 and 2 (super-intensive and intensive olive groves) the price of "irrigated olive trees for olive oil ", to category 3 (almond orchards) the price of "rainfed nut trees", and to category 4 (cereal crops and pastures) the price of "other pasture land" (cereal crops in the study area are not harvested and are used as pasture for sheep and goats).

*2.4. Assessment of the Impacts of the Deployment of PPs on the Biodiversity of the Study Area Based on the Environmental Objectives of the Taxonomy Regulation*

The evaluation of potential damage to Important Conservation Areas (ICAs) (i.e., protected areas, key biodiversity areas, wilderness areas, sensu [27], was approached at two complementary levels:

(i)   considering the Nature 2000 Network and protected areas (according to information from the MITECO Metadata catalogue [28]):

1.   SAC and SPA,
2.   Protected Natural Areas (PNAs).

(ii)  through the study of important areas for the conservation of key species, namely indicator bird species of special interest to the EU, following:

1.   Environmental zoning for renewable energies [29], published in December 2020 (hereafter MITECO zoning).

The five initial categories of environmental sensitivity to PPs were grouped into two groups: (i) maximum, very high and high; (ii) moderate and low.

2.   Methodological guide for the assessment of the repercussions of solar installations on steppe bird species [30] and strategies for the conservation and recovery of endangered species at a state level, specifically those referring to steppe birds [31] (hereafter Steppe birds MITECO).

Both documents use the same geographical criteria, identifying the UTM 10 × 10 km squares with the presence of steppe bird species, based on [32], national censuses of the common bird monitoring programs carried out by Spanish Ornithological Society (SEO/Birdlife) and the latest six-year periodic reports for compliance with Article 12 of the Birds Directive. We consider all grids in which at least one bird species appears.

3.  Environmental zoning, published in January 2021, associated with the guide of the General Directorate of Natural Environment, Biodiversity & Protected Spaces for the analysis of the location of PPs [33] (hereafter Junta de Andalucía zoning).

This guide divides the Andalusian territory into 3 categories. These are: (i) non-compatible zones (critical areas for steppe birds in the annexed zoning of the Environmental Conditions Viewer), in which PPs will be definitively reported unfavorably; (ii) conditioned compatibility zones (areas considered strategic for steppe birds), in which PPs may be installed if the environmental assessment is favorable; (iii) compatible zones (areas with no current or historical presence of steppe birds), where the zone is recommended by the Andalusian Administration for the location of PPs.

4.  Recovery plans for endangered species of the Junta de Andalucía [34] (hereafter Junta de Andalucía Recovery plans).

In our context, there are 2 areas of application for these plans: (i) areas of the scope of action of the Steppe Bird Recovery Program (AASBR, ZAPRAE in Spanish), and (ii) dunes, sandbanks and coastal sandy habitats (hereafter Dunes).

5.  Master Plan for the Improvement of Ecological Connectivity in Andalusia [18] (hereafter Connectivity plan).

This plan defines 4 typologies of territories: landscapes of interest for ecological connectivity (IEC, PIC in Spanish) and priority intervention areas (PIA, API in Spanish) define a system of protected and unprotected spaces capable of channeling a large part of the ecological flows that occur in Andalusia. Reinforcement areas strengthen the functionality of the previous ones. Finally, pilot areas aim to improve spaces that are unfavorable for connectivity due to their current characteristics. In the study area, there are only PIAs and IECs, which we have grouped into a single category, and areas not included in the Connectivity plan.

6.  Important Bird Areas (IBAs) [28].

BirdLife IBAs Inventory provides a list of priority conservation areas for birds in each EU member state to satisfy, among others, the requirements of the Directive 2009/147/EEC on the Conservation of Wild Birds and the declaration of SPAs. IBAs identified by BirdLife have the same value as SPAs declared under Directive 2009/147/EEC, and so the deterioration of these areas as habitats for species covered by that Directive must be avoided.

7.  Priority Habitats of Community Interest (HCIs) [35].

HCIs are natural and semi-natural areas that either are threatened with extinction, have a reduced distribution, or are representative examples of one or more biogeographic regions of the EU. Priority HCIs are those threatened with extinction in the EU. Their description and ecological characterization are included in [36].

*2.5. Data Analysis*

Our data set included PPs in two stages: (i) resolved (either approved or denied) and published in BOE, BOJA or BOP, most of which were evaluated and built during 2017–2020; (ii) in the pipeline, namely PPs whose evaluation started in 2021. In this second category, we distinguished 3 subcategories: with right of access to the power grid or projects exposed to public information, with Environmental Opinion (favorable or unfavorable) and with Binding Report. Our analyses considered all or some of these categories (see below).

These 2 main stages of PPs deployment enabled us to study temporal changes in developers' preference criteria for the type of land used to build PPs, the characteristics of the PPs, and the effectiveness of public allegations, made by expert bodies and associations, which emerged mainly from the end of 2020 onwards.

We georeferenced [37] all the proposed PPs projects (fenced area) since 2017 (n = 35). This information enabled us to estimate their overlap with:

(i)   the 4 land categories mentioned above. Olive groves were georeferenced based on orthophotos, specifically with PNOA maximum actuality [24], and our results were very similar to those obtained by [22]. For this study, we distinguished between resolved PPs (dated January 2017–December 2020) and PPs in the pipeline (announced between January 2021 and September 2022).

(ii)  ICAs. To evaluate the overlap of PPs with Priority HCIs, we used the unique HCI layer available at [35]. This layer represents the basic official cartography on the potential distribution of HCIs in Andalusia. For this study, we distinguished among approved, denied and in the pipeline PPs.

The overlap (i.e., impact) of PPs with ICAs revealed the effectiveness of the process regulating SE deployment in terms of ICA protection. Additionally, we addressed this issue by analyzing the effect of allegations to PPs by experts (namely Estación Experimental de Zonas Áridas EEZA/CSIC and SEO/BirdLife), mainly occurring from the end of 2020 onwards. We distinguished between PPs: (i) without (n = 9), and (ii) with expert allegations (n = 6) (resolved or in progress) when decisions after allegations had been publicized (i.e., Final Decision—Authorization or Denial, PPs with Environmental Opinion, PPs with Binding Report, Modifications of the original project submitted publicly by promoters). For the latter, we compared the original project (i.e., before allegations) and the modified project (after allegations) to explore changes in: (i) overlap between PPs and ICAs, (ii) overlap between PPs and land types, (iii) efficiency of land occupation by calculating the fenced hectares occupied per peak power MW (MWp), and (iv) length of evacuation lines.

Finally, in an attempt to harmonize SE deployment and biodiversity conservation, we explored the possibility of PPs being located in areas currently occupied by olive groves and analyzed the consequences of such a proposal in terms of: (i) efficiency of land occupation by PPs in different land types; (ii) water savings, calculated as the subtraction between the water consumed by the olive groves and the water that would be consumed for washing the solar modules that would occupy the olive groves. Water expenditure for cleaning the modules was estimated, considering 5 $m^3$/MWp and 4 washes per year [38]. Water consumption by olive groves was taken from [22].

Simulations of land use efficiency and water savings were carried out in 2 cases: (i) if all plants (n = 35) were located on land occupied by olive groves, (ii) and if plants in the pipeline (n = 25) were built on land occupied by olive groves.

Sample sizes varied among variables and comparisons due to lack of information in some PPs.

## 3. Results

### 3.1. Characteristics of the Deployment of Solar Energy

The main feature of SE deployment in the study area is the high concentration of PPs due to the existence of electric nodes (Figure 1). There are eight connection points (six power lines with suitable characteristics and 2 substations), but most PPs (28) are connected to the substation Tabernas 220 kv/400 kv, one to the substation La Ribina 400kv, and six to distribution power lines with suitable characteristics (Figure 1). Therefore, the location of the connection points is a key element, with developers seeking proximity to these points. However, a clear temporal trend is evident in a short period. The shortest distance from PPs to their connection points is double in PPs in the pipeline (2021–2022) if compared with resolved PPs (2017–2020) (Table 1), which reflects the saturation of territory around the main connection point (substation Tabernas 220 kv/400 kv). The length of the evacuation lines is not, however, longer (Table 1), because several PPs in the pipeline (16 out of 25) shared their entire line or at least part of it. In contrast, only two of the resolved PPs partially shared the evacuation line.

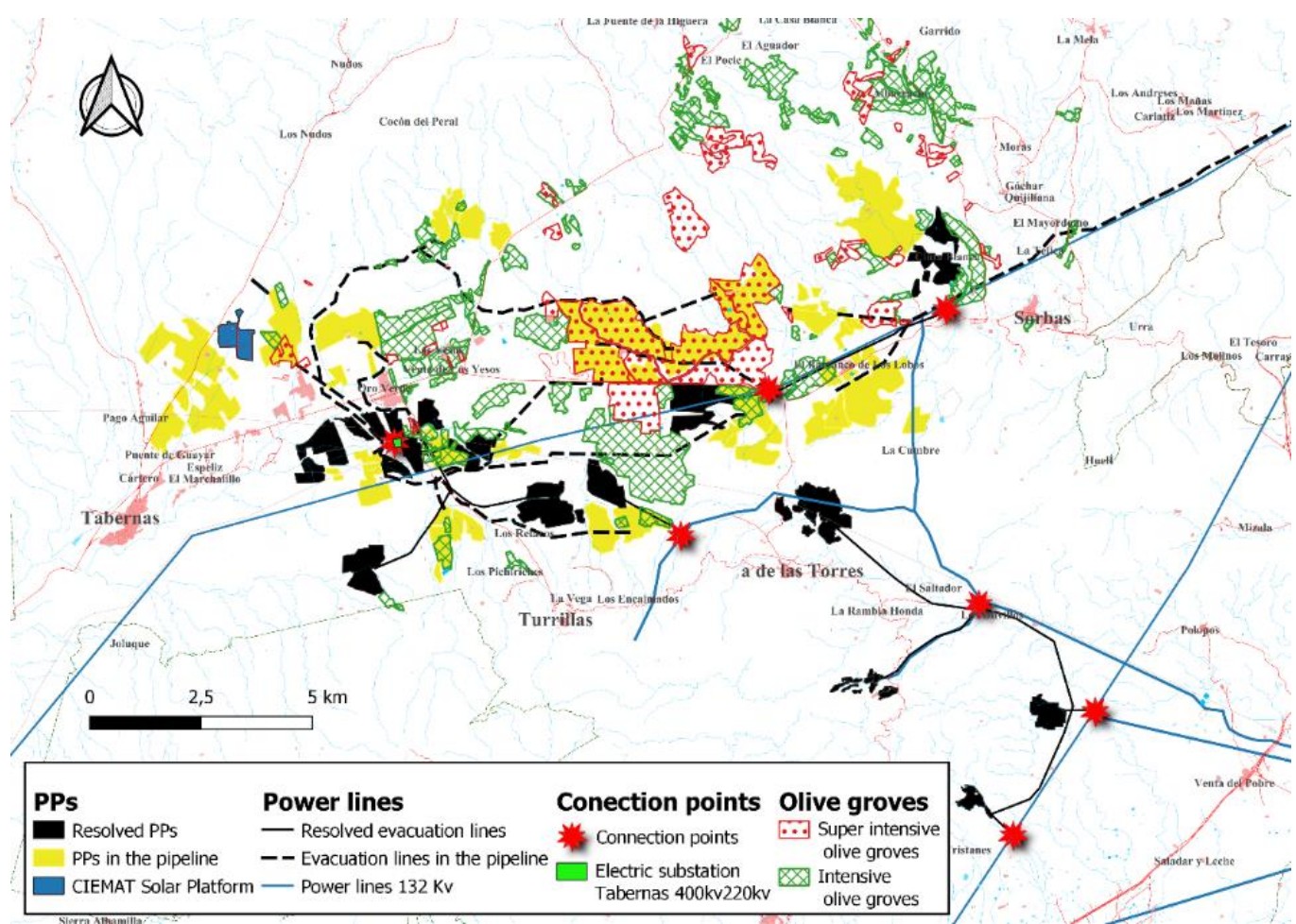

**Figure 1.** PPs and evacuation lines resolved and in the pipeline, connection points, and super-intensive and intensive olive groves. La Ribina substation and the power lines (220 kv and 400 kv) that converge at Tabernas substation are not shown for clarity.

**Table 1.** Main characteristics of the PPs resolved and in the pipeline. Fenced area (ha) of the PPs was calculated with GIS based on information from PP projects. Connection point: pre-existing infrastructure (substation or power line) where the generated electricity is evacuated. Average distance to the connection point: shortest distance from the connection point (132 kv line or 220 kv/400 kv substation) to the nearest point of each PP. Total length evacuation line and Average length evacuation line: based on data provided by the PP projects. Sample size is 35 PPs for all variables except for Total and Average length of the evacuation line, which is 33 PPs.

| PPs | Fenced ha | Average Fenced ha | Peak Power (MWp) | Average Peak Power (MWp) | Average Land Occupation (ha/MW) | Average Distance to Connection Point (m) | Total Length Evacuation Line (m) | Average Length Evacuation Line (m) |
|---|---|---|---|---|---|---|---|---|
| Resolved (2017–2020) (n = 10) | 1093 | 109.3 | 476.6 | 47.7 | 2.29 | 3311.50 | 29,730 | 3303.37 |
| In the pipeline (2021–2022) (n = 25) | 2809 | 112.36 | 1290.3 | 51.6 | 2.18 | 7195.12 | 89,767 | 3740.29 |
| Total | 3902 | 111.49 | 1766.9 | 50.5 | 2.21 | 6085.51 | 119,497 | 3414.18 |

Concerning land type occupancy, resolved ad earlier built PPs avoided super-intensive and intensive olive groves (Table 2). In contrast, PPs in the pipeline occupied such land

type in a proportion as high as 32.1%. This means an increase in the price of land paid by PPs in the pipeline (average price: 16,514.9 €/ha) in comparison to the one paid by resolved PPs (average price: 4048.07 €/ha) (Table 2).

**Table 2.** Occupation of the main land types by PPs (resolved, in the pipeline and total) and price of each land category. The original crop types were aggregated into the 4 types shown in the table (see Methods Section 2). To estimate the value of each land type, the most frequent price reported by Junta de Andalucía (2021a) was used (see Methods Section 2). For olive groves, the price of "Irrigated olive trees for olive oil" was considered (42,320 €/ha). For almond groves, the price of "Rainfed nut trees" was considered (6799 €/ha), and for rainfed cereal crops and grasslands the price of "Other pasture land" was considered (3561 €/ha). The price shown is the value per ha per number of ha occupied.

| PPs | Fenced ha | Super-Intensive Olive Groves (%) | Intensive Olive Groves (%) | Almond Orchards (%) | Rainfed Cereal Crops & Grasslands (%) |
|---|---|---|---|---|---|
| Resolved (2017–2020) (n = 10) (ha) | 1093 | 0.0 (0.0) | 2.9 (0.3) | 129.7 (11.9) | 960.4 (87.9) |
| Land prices (€) occupied by resolved PPs | | 0 | 122,728.0 | 881,830.3 | 3,419,984.4 |
| In the pipeline (2021–2022) (n = 25) (ha) | 2809 | 702.0 (24.9) | 201.1 (7.2) | 427.5 (15.2) | 1478.4 (52.6) |
| Land prices (€) occupied by PPs in the pipeline | | 29,708,640.0 | 8,510,552.0 | 2,906,572.5 | 5,264,582.4 |
| Total (2017–2022) (n = 35) (ha) | 3902 | 702.0 (17.9) | 204.0 (5.2) | 557.2 (14.3) | 2438.8 (62.5) |
| Total Land prices (€) | | 29,708,640.0 | 8,633,280.0 | 3,788,402.8 | 8,684,566.8 |

### 3.2. Impact of PPs on Important Areas for Conservation (ICAs)

No PPs have been installed or planned in Natura 2000 Network areas (SACs, SPAs) nor in PNAs.

However, the approved PPs have a significant impact on the ICAs (Table 3, Figure 2). For example, 39% of the area occupied by approved PPs overlaps with maximum–high environmental sensitivity areas (Category 1, MITECO zoning), and ca. 41% is within areas that Junta de Andalucía itself considered as "Not Compatible" since they are critical for steppe birds. Birds (namely steppe ones) are seemingly the main casualties of PPs, evidenced by the high percentage of favorable habitat (e.g., the AASBR "Campo de Tabernas-Sierra de Alhamilla" and IBAs) occupied by the latter (Figure 2). High-priority habitats and key ecological features of the area are also widely affected (e.g., all approved PPs are within important areas for the maintenance of the ecological flows) (Table 3).

PPs in the pipeline involve greater impact to ICAs if compared with approved PPs, except in the case of IBAs, where the overlap drops from 89% to 55% (Table 3).

Strikingly, only ca. 4% of all the PPs (approved and in the pipeline) are located within what the Junta de Andalucía itself considers a "Compatible Zone" (Table 3).

The only PP definitively denied had a large area occupying highly sensitive zones, namely important areas for steppe birds such as the protected Dupont lark (*Chersophilus duponti*) (Table 3, Figure 2a).

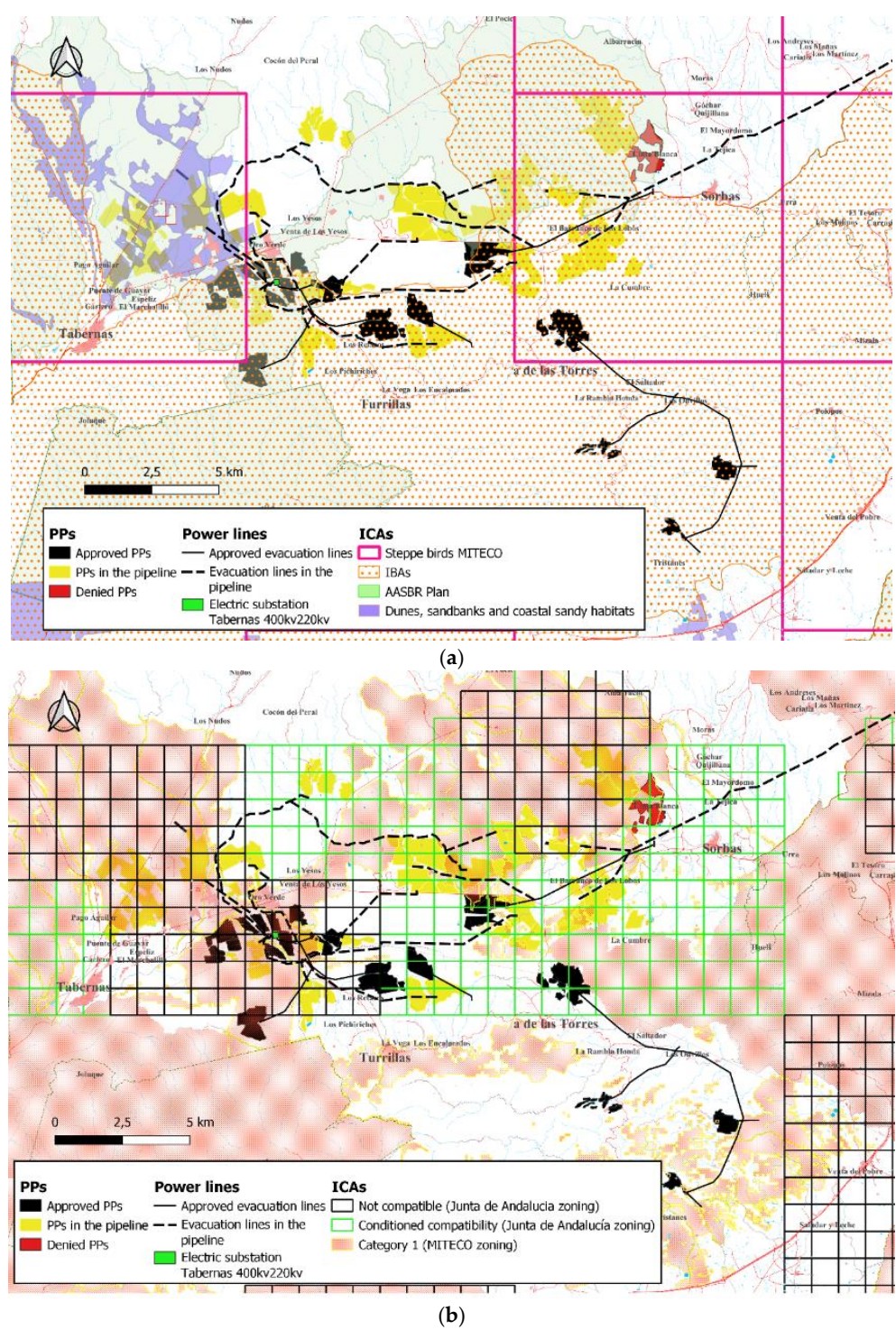

**Figure 2.** Overlap of the approved, denied and PPs in the pipeline with ICAs. (**a**) Guide for the assessment of the repercussions of solar installations on steppe bird species (MITECO 2021) and strategies for the conservation and recovery of endangered species (MITECO 2022c), recovery plans for endangered species of the Junta de Andalucía, namely AASBR and Dunes and coastal sandy habitats (Junta de Andalucía 2022a), and IBAs. (**b**) Zoning carried out by MITECO and Junta de Andalucía. Neither the Plan for the Improvement of Ecological Connectivity nor the HCIs are represented since the overlap with PPs is very high (100% and 67.7% respectively) and would make it difficult to understand the figure.

### 3.3. Involvement of Experts and Academics during the Public Consultation Phase: Allegations to PP Projects

The first 9 plants proposed (during 2017–2020) did not receive expert allegations and were authorized, despite their considerable impact on the various ICAs (Table 3, Figure 3).

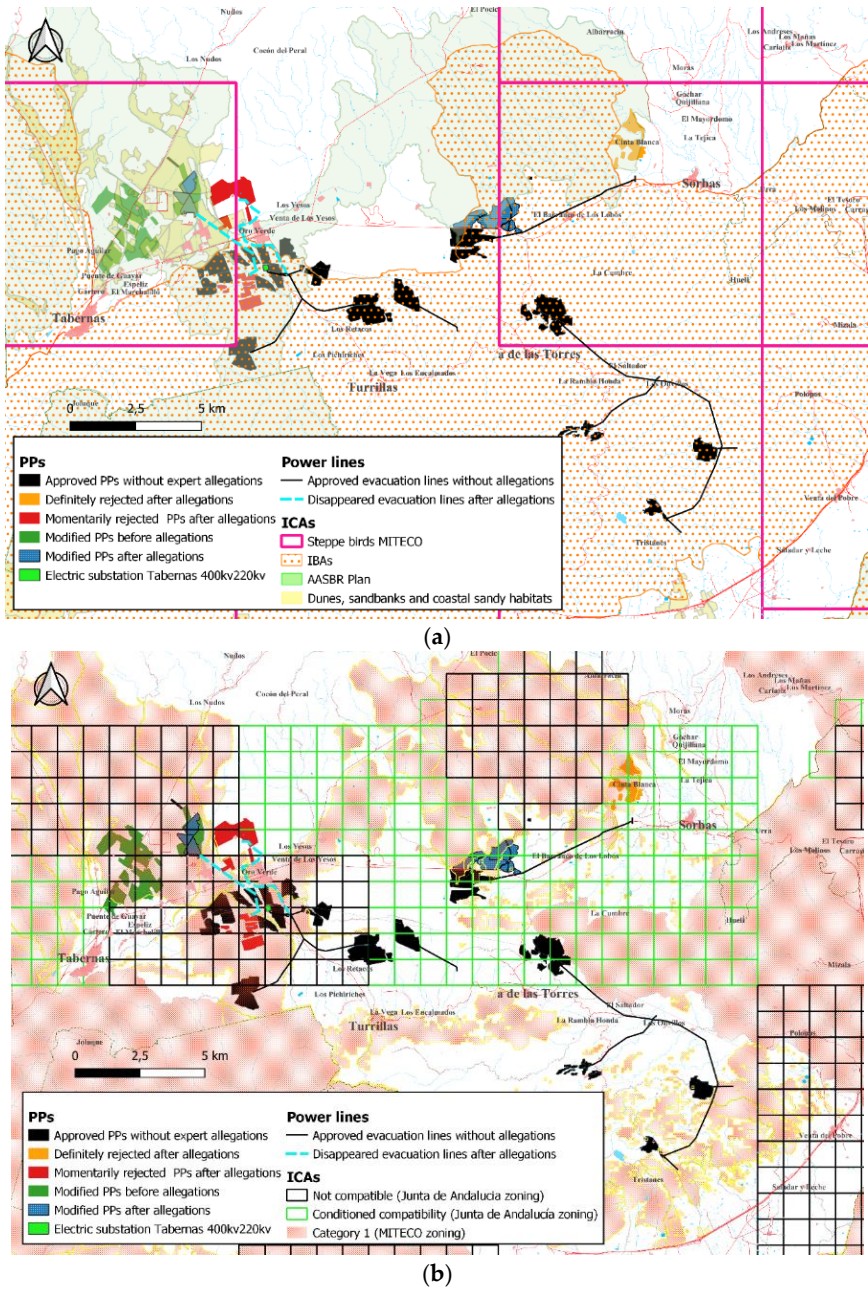

**Figure 3.** Overlapping between ICAs and approved PPs without expert allegations, definitely rejected PPs, momentarily rejected PPs after expert allegations, and modified PPs before (original location) and after expert allegations. Approved evacuation lines and evacuation lines rejected after allegations. (**a**) Guide for the assessment of the repercussions of solar installations on steppe bird species (MITECO 2021) and strategies for the conservation and recovery of endangered species (MITECO 2022c), recovery plans for endangered species of the Junta de Andalucía, namely AASBR and Dunes and coastal sandy habitats (Junta de Andalucía 2022a), and IBAs. (**b**) Zoning carried out by MITECO and Junta de Andalucía. Neither the Plan for the Improvement of Ecological Connectivity nor the HCIs are represented since the overlap with PPs is very high (100% and 67.7% respectively) and would make it difficult to understand the figure.

Our review of the process followed for six plants that received expert allegations reveals that the latter had 2 different consequences. On the one hand, three plants were considered environmentally unfeasible by the Administration: one is definitively denied (Table 3, Figure 2), another has received until date an unfavorable Binding Report and another has until date an unfavorable Environmental Judgment. On the other hand, in 3 cases the developers modified their initial projects, fundamentally changing the location and design of key aspects (e.g., the evacuation lines). The result of both types of consequences is shown in Table 4 and Figure 3. Specifically, the refusal and modification of PPs entailed, compared to the original projects, a lower impact (ranging 75–91%) on the most sensitive ICAs except for the IBA (Table 4) since PPs were moved to land occupied by olive groves (see below). The greater impact on IBAs is explained by their degradation in the last decade, when a significant area was transformed into olive groves. In fact, all the IBAs affected by the modified PPs after the allegations are actually intensive (635 ha) and super-intensive (849 ha) olive groves. Recent degradation also explains the light decrease in the overlapping of modified PPs with some ICAs (Category 1, MITECO zoning and AASBR).

Concerning occupation of land types, prior to allegations, the PPs initially tended to avoid olive groves (only ca. 5% of their area laid on land previously occupied by this crop, Table 5) in favor of cheaper land types. However, after expert allegations, 3 of the PPs shifted to olive groves (Figure 3), so that ca. 88% of their area is located on such land type (namely super-intensive olive groves, modifications representing an increase in 571% over the initial occupation of this cultivation).

An additional effect, related to the displacement of PPs to olive groves, is an increase in the efficiency of land occupation. Despite a reduction in ca. 72% in the area to be occupied, the reduction in power is only 50% (Table 5). This is because olive groves are located in flatter areas than natural and semi-natural habitats, which means that less hectares are needed per MW (i.e., equivalent to a reduction in ca. 45%, Table 5).

Finally, in the resolved PPs without expert allegations, none of the evacuation lines were buried, regardless of whether they crossed ICAs or not. As a result, 29730 m of overhead power lines were built (Table 1). However, in the PPs that received allegations, no overhead evacuation line was maintained, since the project was denied (n = 3). Alternatively, in the three cases in which the location of the PPs changed, evacuation lines were buried up to the connection node or to other overhead evacuation lines of nearby projected PPs (Figure 3).

*3.4. Proposals for Solar Energy Deployment Compatible with Biodiversity Conservation: Replacing Olive Groves with PPs*

The area of (intensive and super-intensive) olive groves available allows the installation of 2182.2 MW (Table 6), amply sufficient to install all the PPs initially planned in the area (1766.9 MW, Table 1).

**Table 3.** Overlapping of PPs (approved, denied, in the pipeline and all of them) with different types of ICAs. Data are given in ha and percentages (in brackets). ICAs are as follow: 1.—MITECO zoning: Category 1: Maximum, very high and high environmental sensitivity zones. Category 2: Moderate and low environmental sensitivity zones. 2.—Methodological guide for the assessment of the repercussions of solar installations on steppe bird species (MITECO 2021) and the strategies for the conservation and recovery of endangered species (MITECO 2022c): These two ICAs have been grouped into a single category ("Steppe birds MITECO") because they use the same geographical definition. 3.—Junta de Andalucía Zoning: The original zoning of the guide is shown. 4.—Junta de Andalucía Recovery Plans: shown the areas of both plans (Dunes, sandbanks and coastal cliffs and AASBR). 5.—Junta de Andalucía Connectivity Plan: PIAs & IECs have been grouped together. 6.—Important Bird Areas (IBA). 7.—Priority HCIs: All priority HCIs are considered together, based on the REDIAM single layer. For more details see Methods.

| Type PPs | Fenced ha | MITECO Zoning ha (%) | | Steppe Birds MITECO ha (%) | Junta de Andalucía Zoning ha (%) | | | Junta de Andalucía Recovery Plans ha (%) | | Connectivity Plan PIA & IEC ha (%) | IBAs ha (%) | Priority HCIs ha (%) |
|---|---|---|---|---|---|---|---|---|---|---|---|---|
| | | Category 1 | Category 2 | | Not Compatible | Conditioned Compatibility | Compatible | Dunes | AASBR | | | |
| Approved (n = 9) | 975 | 378.9 (38.9) | 596.1 (61.1) | 114.3 (11.7) | 396.1 (40.6) | 418.8 (42.9) | 160.1 (16.4) | 0.0 (0.0) | 363.5 (37.3) | 975.0 (100.0) | 877.0 (89.9) | 512.1 (52.5) |
| Denied (n = 1) | 118 | 110.1 (93.3) | 7.9 (6.7) | 118.0 (100.0) | 0.0 (0.0) | 118.0 (100.0) | 0.0 (0.0) | 0.0 (0.0) | 109.7 (92.9) | 118.0 (100.0) | 0.0 (0.0) | 98.6 (83.6) |
| In the pipeline (n = 25) | 2809 | 1209.6 (43.1) | 1599.4 (56.9) | 1463.6 (52.1) | 1270.8 (45.2) | 1523.5 (54.2) | 14.7 (0.5) | 279.4 (9.9) | 1368.8 (48.7) | 2809.0 (100.0) | 1558.0 (55.5) | 2030.5 (72.3) |
| Total (n = 35) | 3902 | 1698.6 (43.5) | 2203.4 (56.5) | 1862.9 (47.7) | 1666.9 (42.7) | 2060.3 (52.8) | 174.8 (4.5) | 279.4 (7.2) | 1842.0 (47.2) | 3902.0 (100.0) | 2435.0 (62.4) | 2641.2 (67.7) |

**Table 4.** Effects of allegations on the impact of PPs on ICAs. The overlapping of ICAs and PPs that received allegations (original proposals) and the modified projects after allegations is shown. Data are given in ha and percentages (in brackets). The last row shows (as a %) the total effect of allegations (denial and modifications of PPs) by referring the difference between the areas occupied by the original and the modified projects to the area initially proposed. The sign means an increase (+) or a decrease (-) in overlap between PPs and ICAs with regard to the original project. For a description of ICAs see Table 3 and Methods.

| PPs | Fenced ha | MITECO Zoning ha (%) | | Steppe Birds MITECO ha (%) | Junta de Andalucía Zoning ha (%) | | | Junta de Andalucía Recovery Plans ha (%) | | Connectivity Plan PIA & IEC ha (%) | IBAs ha (%) | Priority HCIs ha (%) |
|---|---|---|---|---|---|---|---|---|---|---|---|---|
| | | Category 1 | Category 2 | | Not Compatible | Conditioned Compatibility | Compatible | Dunes | AASBR | | | |
| With allegations (original Project) (n = 6) | 840 | 660.0 (78.6) | 180.0 (21.4) | 697.5 (83.0) | 653.1 (77.7) | 186.9 (22.2) | 0.0 (0.0) | 279.4 (33.3) | 679.0 (80.8) | 840.0 (100.0) | 79.0 (9.4) | 703.4 (83.7) |
| Modified projects after allegations (n = 3) | 232 | 161.5 (69.6) | 70.5 (30.4) | 60.6 (26.1) | 72.0 (31.0) | 160.0 (69.0) | 0.0 (0.0) | 39.2 (16.9) | 165.5 (71.3) | 148.0 (63.8) | 152.5 (65.7) | 175.2 (75.5) |
| Changes after allegations (%) | −72.4 | −75.5 | −60.8 | −91.3 | −89.0 | −14.3 | 0.0 | −86.0 | −75.6 | −82.4 | +93.0 | −75.1 |

**Table 5.** Effects of allegations on the impact of PPs on land occupation. The overlapping of land previously occupied by olive orchards and PPs that received allegations (original proposals) and the modified projects after allegations is shown. Data are given in ha and percentages (in brackets). Data on fenced area, peak power and the efficiency of land occupation (ha/MW) is also shown. The last row shows (as a %) the total effect of allegations (denial and modifications) by referring the difference between the values of the original and the modified projects to the value initially proposed. The sign means an increase (+) or a decrease (-) in each variable with regard to the original project.

| PPs | Fenced ha | Peak Power (MWp) | Land Occupation (ha/MW) | Super Intensive Olive Orchards ha (%) | Intensive Olive Orchards ha (%) |
|---|---|---|---|---|---|
| With allegations (original Project) (n = 6) | 840 | 300 | 2.8 | 28.0 (3.3) | 15.7 (1.9) |
| Modified projects after allegations (n = 3) | 232 | 150 | 1.55 | 188.0 (81.0) | 15.7 (6.8) |
| Changes after allegations (%) | −72.38 | −50 | −44.6 | +571.4 | 0 |

**Table 6.** Potential of olive groves in the study area to harbor PPs. Original area of olive groves, area of olive groves occupied by approved PPs and PPs in the pipeline, area of olive groves still available, average land occupation of PPs in olive groves (ha/MW) and power that can be installed in olive groves (assuming 1.53 ha/MW for all cases). n/d: data not available.

| | Olive Orchards (ha) | Approved PPs (ha) (n = 9) | PPs in the Pipeline (ha) (n = 25) | Available Olive Orchards (ha) | Average Land Occupation by PPs (ha/MW) | Potential Power (MW) |
|---|---|---|---|---|---|---|
| Super-intensive | 1600.7 | 0.0 | 890.0 | 710.7 | 1.53 | 464.5 |
| Intensive | 2832.0 | 0.0 | 204.0 | 2628.0 | n/d | 1717.6 |
| Total | 4432.7 | 0.0 | 1094.0 | 3338.7 | - | 2182.1 |

Such a proposal has important consequences in terms of land occupancy and water saving.

The efficiency of soil occupation by PPs is higher in PPs located on land previously occupied by olive groves (1.53 ha/MW, n = 13 PPs) than on land occupied by other crops or natural vegetation (2.43 ha/MW, n = 22 PPs). If all plants (3902 ha, 1766.9 MW, n = 35, Table 1) had been initially planned on land occupied by olive groves, 30.7% less area would have been occupied (2703.4 ha). If the PPs in process (2809 ha, 1290.3 MW, Table 1) were placed on olive groves, 29.7% less land would be occupied (1974.2 ha).

Washing the modules of all the planned PPs (1766.9 MW, n = 35, Table 1) requires 0.035 hm$^3$/year (assuming 20 m$^3$ of pure water/MW year). Water consumption of super-intensive and intensive olive groves in the area is 4480–6590 m$^3$/has and 2460–3460 m$^3$/ha, respectively [22]. Making a conservative calculation, if all the plants (3902 ha, n = 35, Table 1) had been initially planned on land occupied by intensive olive groves (which consumes the least water), 9.6–13.5 hm$^3$ would have been saved. If the plants in the pipeline (2809 ha, Table 1) were built on available super-intensive (1600.7 ha) and (the rest) on intensive (1208.3 ha) olive groves (Table 6), considering the abovementioned irrigation water expenditure ranges and flushing water expenditures, 10.1–14.7 hm$^3$ would be saved.

## 4. Discussion

This is, to our knowledge, the first study on solar energy deployment at a landscape scale that considers its environmental impact, as well as the legal and land management implications. The main feature of SE deployment in the study area is the high concentration of PPs in a small extent, due to the existence of electric nodes. Therefore, the location of the connection points is a key element that should be dealt with in a territorial energy plan with strategic environmental assessment. A second important factor is the occurrence of cheap land around such points. These factors foster the development of PPs, regardless the conservation or landscape value of the areas they occupy. Yet, saturation occurs over time and PPs have to move progressively away from the connection points.

This saturation process, accompanied by the effect of expert claims (see below), evidences two advantageous realities: (i) to save costs, the companies agreed to share their evacuation lines, something that did not happen in the case of the first PPs built; (ii) although initially companies looked for the cheapest soils (e.g., rainfed crops and pastures) and avoided installing PPs on olive groves, the cost of occupying the latter land type is not a limiting factor, as evidenced by the continuous projects being developed in this type of crop.

Another important feature of SE deployment in the study area is its high environmental costs. While we found no direct impact (i.e., occupancy) on Natura 2000 network, indirect effects are likely [39] since many Natura 2000 sites have been declared as suffering from such issues because of the existence of highly mobile organisms (e.g., birds or bats) that may frequently use areas outside the Natura 2000 network. Moreover, other areas considered valuable for biodiversity by national and regional administrations, are seriously affected by the PPs. In our case, wildlife inhabiting those habitats (namely steppe birds and raptors) become major targets of such infrastructures (see also [6]). In fact, the occupation of areas considered appropriate by the Administrations seems to be the exception rather than the rule. In our case study, this is due to the scarcity of compatible zones around the main connection points. Moreover, only 683 ha (3.8%) out of 17,925 ha considered as compatible zone do not overlap with ICAs. This evidences: (i) that in our study area, olive groves are the land type on which the deployment of PPs will have the least impact, and (ii) the existence of contradictions between the real environmental value of the various land types and the value officially assigned to them due to lack of updated and rigorous information (see below).

It could be argued that some of the initial PPs were approved before guidelines for their location were published by the Administration (e.g., Junta de Andalucía Guide, see Methods). However, data for the design of the zoning associated with these guides are based on information on species (censuses, distribution reports of endangered species) and

important areas (e.g., AASBR, IBA) recorded and owned by the Administration. Therefore, at least in some cases, the Administration had the necessary information to make the right decision to avoid damage to environmentally sensitive areas.

Our results show that the administrative procedure for evaluating (and eventually approving) a PP does not guarantee a low environmental impact at all. Several reasons may explain this. First, developers very often conduct deficient EISs, based on little field sampling effort and neglecting the impact of projects on wildlife, habitats and landscapes.

Second, public Administrations admittedly lack detailed and updated information on the distribution and abundance of endangered species and on the value of some spaces in spite of the legal obligation (arts. 4, 6 and 18 Directive 92/43 and art. 4 and 10 Directive 2009/147, Text S1). This causes some ICAs to be poorly defined. For instance, it is recognized that HCIs may be overrepresented [40]. Therefore, it is necessary that both the promoters and the Administration use an adequate methodology to confirm and identify the presence of the priority HCIs in those sites. Something similar occurs with other ICAs such as AASBR and IBAs (areas declared as IBA were occupied long ago by olive groves, so that their substitution by PPs not only has no serious environmental impact but is desirable). This, in turn, causes other criteria to be poorly designed (such as MITECO's sensitivity zoning, which is based on these ICAs). There are also some contradictions between criteria followed by different administrations. For example, the only PP denied was located in a highly sensitive area according to MITECO, but not according to the Junta de Andalucía (see Table 3). This lack of information on the distribution of vulnerable species is especially worrying, especially taking into account that recent legislative changes, through with specific exceptions, reduce the guarantees of biodiversity protection and environmental public participation achieved after many years of scientific knowledge, environmental awareness and activism (Royal Decree act 11/2022 and legislative proposals of the European Commission COM(2022)222 final and COM(2022)591 final, Text S1). It is important to point out that this outdated information also affects events in the opposite direction, since important areas for steppe birds are outside IBAs and AASBR (own data). Assessing the impact of renewables on biodiversity just on the basis of ICAs is not correct, given that many species spend a large part of their cycle outside ICAs (e.g., [41]). Moreover, it has been demonstrated that the EU Biodiversity Strategy, based on the protection of SPAs, is not effective for the conservation of threatened steppe birds within these Natura 2000 sites [42] since they do not consider that many areas of high conservation for specific taxa (e.g., steppe birds) occupy agricultural areas [6].

Third, the lack of staff and time to carry out a proper environmental impact assessment, especially given the flood of projects in a short time, can undermine the effectiveness of the Administration's assessment.

There are several consequences of failures in the regulatory procedure for the deployment of renewables. We found that the approval of PPs had a call effect, in spite of serious impact on ICAs, as evidenced by the fact that PPs in the pipeline have an even greater impact than those initially approved. Moreover, approval of PPs in spite of the impact described here, involves significant infringements of the legislation for evaluating environmental impacts (Act 21-213 on environmental assessment and Directive 2011/92, Text S1) and protection of biodiversity (Act 42-2007 on natural heritage and biodiversity, Directive 92/43 and Directive 147/2009, Text S1), the most important of which are: (i) failure to comply with the precautionary principle (art. 191.2 Treaty on the Functioning of the European Union); (ii) failure to take into account the best available scientific knowledge; (iii) the failure to evaluate the synergistic effect.

Importantly, our study reveals that the experts' allegations can contribute to mitigating the shortcomings of the regulatory procedure. PPs that did not receive expert allegations were authorized despite their considerable impact on the various ICAs (Table 4, Figure 3). Apparently, the experts' allegations provided information and arguments to the Administration, with which they could reject (at least momentarily) several PPs and force the companies to modify the initial projects by relocating to olive groves, thus reducing their

environmental impact. Such modifications also evidenced that the land use efficiency of the PPs is higher in olive groves than in the areas initially chosen (see below) and that the length of the evacuation lines, which are known to have a negative impact on many bird species [43,44], is reduced. Yet, current changes in environmental regulations (Royal Decree 11/2022 and amendments of Directive (EU) 2018/2001 with COM(2022)222 final) aim to exclude renewable energies from the environmental impact and public information procedure, replacing it with a process of granting express authorizations, based on environmental zoning designed by the Administrations.

Such changes do not appear to be compatible with EU law because: (i) they violate the principle of non-regression, by excluding projects from the environmental impact assessment that were under the environmental directives (art. 2 and 4 Directive 2011/92 and 6(3) Directive 92/43, Text S1), without there being an environmental cause for doing so; (ii) they represent a regression in the right to participation (art. 6 Aarhus Convention, art. 6 Directive 2011/92 and art. 6(3) Directive 92/43, Text S1); (iii) they jeopardize biodiversity conservation because, as shown here, proactive allegations (i.e., providing reasonable alternatives) help to alleviate the lack of reliable and updated information on the distribution and status of endangered species. Such shortcomings translate into poorly designed environmental zoning by administrations. The latter is a general problem already detected by the EU Biodiversity Strategy to 2030, in particular on the conservation status of species and areas of community interest [45].

Fortunately, in Campo de Tabernas there is a win–win alternative to balance the deployment of PPs and the conservation of both biodiversity-sensitive areas and landscape. We propose building the PPs on land previously occupied by super-intensive and intensively irrigated olive orchards that have spread in recent decades, spoiling the aquifer of the Tabernas–Sorbas basin [22]. Such a proposal has clear advantages in terms of multiple areas. The first (i) is land occupation efficiency. Land use intensity is an important impact of PPs [8], even more so in the study area, whose landscape is unique in Europe and considered to be of cultural interest [46]. Since the land occupied by olive groves has been previously flattened [22] (see Martínez-Valderrama et al. 2020), land occupation efficiency of PPs on previous olive groves is higher than the one of PPs elsewhere. The second (ii) is biodiversity conservation. Our study shows that the area currently occupied by olive groves is sufficient to accommodate all the PPs projected. If the location of the PPs had been previously planned on land occupied by olive groves, the destruction of 975 ha of ICA by the already approved PPs could have been avoided (Table 3). The third (iii) is water saving. Water consumption in intensive and super-intensive olive groves in the region ranges 14–20 hm$^3$ [22] (Martínez-Valderrama et al. 2020), which puts the aquifer of the area at risk, and which is producing an accelerated desertification process [22] (Martínez-Valderrama et al. 2020). If the PP in the pipeline were placed in olive groves, at least 10.1–14.7 hm$^3$ per year could be saved, which would allow the recovery of the aquifer since its annual recharge is 5 hm$^3$ per year [22] (Martínez-Valderrama et al. 2020). For this water saving to be really effective, it is necessary to accredit the existence of registered water use rights, and the Binding Report of the PPs (if positive) should imply the cancellation of the irrigation rights of these plots, except for what is strictly necessary for the cleaning of the modules.

*The Imperative Need to Carry Out Prior Strategic Environmental Studies*

The deployment of renewables in our study area has been carried out in the absence of any plan that considers important issues such as biodiversity conservation or landscape. The first aim of such planning should have been the choice of the most suitable sites for the electric nodes (i.e., electric substations). An Electricity and Gas Sector Planning 2008–2016 was approved in 2008 [47], but the effects of future PPs were not taken into account, nor was strategic environmental assessment (Act 6/2009 and Directive 2001/42, Text S1) completed. In the absence of this plan, the deployment of PPs around the connection points and the design of the evacuation lines (sharing between the different projects and developers) had (and has) to be carefully regulated. If so, the negative impact of PPs shown in this study

had been lower. In the study area, such prior planning (i.e., Strategic Territorial Plan) would have been very easy to design, only replacing olive groves by PPs.

To our knowledge, the procedure being used in the study area (evaluation of each PP without a previous general study) constitutes an infringement by omission of the obligation of strategic environmental assessment (arts. 1 and 3 Directive 2001/42, Text S1) as the implementation of connection nodes has been planned without strategically assessing their effects. Moreover, this implies legal uncertainty for both the Administration and the developers, which is not desirable in any case.

The specialized literature emphasizes that the accumulation of impacts can be much better assessed at a regional or strategic level [48–50]. The accumulation of isolated project decisions without a global perspective can thus become a serious threat to natural heritage, in what has been termed the 'tyranny of small decisions' [51,52].

## 5. Conclusions

The current model of PPs deployment in our study area does not meet the requirements given by the Taxonomy Regulation with regard to protection of biodiversity and ecosystems, and therefore, does not qualify as sustainable economic activity.

To what extent can this pattern be extrapolated to other areas of Spain? Unfortunately, the answer is that SE deployment is seemingly following a similar pattern in the rest of the country. Both experts (e.g., Serrano et al., 2020) and civil demonstrations (namely by rural populations) have warned about the effects of rapid implementation of renewables on biodiversity, ecological connectivity, landscape and rural depopulation.

The implementation of a Strategic Territorial Plan, together with the collaboration of experts (which requires that the procedure include a period of public participation), could make SE an environmentally sustainable activity. It is critical to improve and update information on the distribution and status of vulnerable and threatened species, which requires time and resources. The lack of such information undermines the effectiveness of the proposed regulatory changes in the environmental impact assessment procedure for renewable energies.

Together with the development of the Strategic Territorial Plan, we consider it necessary to solve the contradictions between the criteria used by the various administrations to draw up zoning plans, and the adoption of a uniform methodology in the assessment of EIS. Additionally, we assert that the environmental zoning developed by the administrations should be binding to compel the companies to comply with the requirement of ensuring the least possible environmental impact (art. 21.2 Act 7/2021 on climate change and energy transition, Text S1).

Based on our results, the good news is that there are possibilities to harmonize the deployment of renewables and biodiversity conservation. The uncertainty is whether the Administration will consider such a possibility and implement it.

**Supplementary Materials:** The following supporting information can be downloaded at: https://www.mdpi.com/article/10.3390/land11122330/s1: Table S1: Information on the photovoltaic plant projects (PPs) reviewed in this study; Text S1: Main legislation cited.

**Author Contributions:** Conceptualization, all authors; Methodology, F.V. and L.B.; Software, L.B.; Formal Analysis, L.B. and F.V.; Investigation, all authors; Resources, all authors; Writing—Original Draft Preparation, F.V. and L.B.; Writing—Review & Editing, all authors; Visualization, F.V., L.B., A.L.C. and E.M.; Supervision, F.V.; Project Administration, F.V.; Funding Acquisition, E.M. and F.V. All authors have read and agreed to the published version of the manuscript.

**Funding:** This paper is part of the project TED2021-130035B-100, funded by MCIN/AEI/10.13039/501100011033 and by the European Union "NextGenerationEU"/PRTR.

**Data Availability Statement:** No new data were created or analyzed in this study. Data sharing is not applicable to this article.

**Acknowledgments:** We thank Jesús Veiga, Teresa Martínez, Javier Pérez, Teresa Abaigar and Jesús Benzal for their help with fieldwork and data search. Junta de Andalucía kindly provided information about the environmental impact studies. We are thankful to an anonymous referee for constructive comments and to the inhabitants of Campo de Tabernas for their assistance and support, particularly to the families Castillo-Zamora (Vellsam Materias Bioactivas), López Segura-Escudero, Martínez-Escudero and González-Gómiz.

**Conflicts of Interest:** The authors declare no conflict of interest.

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
