# Peer review of "Deployment of Solar Energy at the Expense of Conservation Sensitive Areas Precludes Its Classification as an Environmentally Sustainable Activity"

_land, doi:10.3390/land11122330_

Round 1

Reviewer 1 Report

The work presented is very relevant taking in consideration that there is a high investment interest in the implementation of PSE, specially in regions with high solar irradiation. The work focus on one specific area in Spain but the methodology adopted could be applied to other regions and other countries.

A suggestion for improvement of the work presented refers to the fact that some of the aspects referred by authors in 4.Discussion should be referred also in the conclusions, namely:

a) need to solve contradictions between criteria followed by different administrations on zoning;

b) adoption of a uniform methodology in the assessment of EIS (Environmental Impact Study).

In the present work it is shown that some of the projects in the pipeline were altered to comply with suggestions received in public consultation. The main change was  the relocation of the PP to olive groves, but no relocation to Compatible zones (Junta de Andalucia zoning). It would be interesting to know what would be the impact for those PP to move to those areas.

The work would benefit from improved format of Tables 1, 2, 3 and 4, by using smaller size letters or using landscape orientation for the Tables.

In some sentences, parenthesis are missing, e.g.:

Page 3 - line 136-137

Page 4 - line 156-157

Page 5 - line 207

Page 16 - line 574

Author Response

> The work presented is very relevant taking in consideration that there is a high investment interest in the implementation of PSE, specially in regions with high solar irradiation. The work focus on one specific area in Spain but the methodology adopted could be applied to other regions and other countries.

Thank you for your kind comments.

> A suggestion for improvement of the work presented refers to the fact that some of the aspects referred by authors in 4.Discussion should be referred also in the conclusions, namely:

  1. a) need to solve contradictions between criteria followed by different administrations on zoning;
  2. b) adoption of a uniform methodology in the assessment of EIS (Environmental Impact Study).

We fully agree with the referee in that these aspects are worth to be included in the conclusion. For instance, the zoning done by the regional Administration (Junta de Andalucía) considers the distribution of endangered species (e.g., steppe birds) and their habitats whereas the zoning done by the national Administration considers the various ICAs. We have added a comment in the new version of the manuscript, but for the sake of brevity, we have summarized the point. Please, see lines 667-669.

> In the present work it is shown that some of the projects in the pipeline were altered to comply with suggestions received in public consultation. The main change was  the relocation of the PP to olive groves, but no relocation to Compatible zones (Junta de Andalucia zoning). It would be interesting to know what would be the impact for those PP to move to those areas.

Again, the referee is right. This is an important issue we have not discussed. This is partly because, as stated in the manuscript, in our case study, Compatible zones around the main connection points are scarce. Specifically, only ca. 26% (17.925 has) out of 69,000 has (area occupied by the main municipalities in our study area) are designed as Compatible zones by Junta de Andalucía. But only 683 has (3.8%) out of those 17925 has do not overlap with ICAs. This evidences: i) that in our study area, olive groves are the land type where the deployment of solar power plants will have the least impact, ii) the contradictions between the real value of the various land types and the value officially assigned to them due to lack of updated and rigorous information. Thus, a strategic territorial planning and expert advice is badly needed. This issue is now included in the new version of the manuscript (please, see lines 530-534).

> The work would benefit from improved format of Tables 1, 2, 3 and 4, by using smaller size letters or using landscape orientation for the Tables.

We agree with the referee. Since the journal accepts free format submission, we prepared the tables in a “free format”. We are ready to modify them following the editor’s instructions.

> In some sentences, parenthesis are missing, e.g.:

Page 3 - line 136-137

Page 4 - line 156-157

Page 5 - line 207

Page 16 - line 574

Corrected. Thanks for your expert eye.

Reviewer 2 Report

This paper focused Deployment of solar energy at the expense of conservation sensitive areas precludes its classification as an environmentally sustainable activity. The subject matter of this manuscript fits the journal's scope, and the information included in the manuscript seems not to have been published in any other publication so far. However, it seems difficult to adequately evaluate the value of this study because the explanation of the significance of the study, the description of the interpretation and usefulness of the results obtained by the analysis, and the explanation of the model are insufficient. I would like to ask the authors to consider responding to the following comments:

(1)    Would you explicitly specify the novelty of your work? What progress against the most recent state-of-the-art similar studies was made?

(2)    The Introduction section should be improved. It should be dedicated to presenting a critical analysis of state-of-the-art related work to justify the study's objective. In addition, critical comments should be made on the results of the cited works.

(3)    The main objective of the work must be written in a more precise and concise way at the end of the introduction section. Please carefully check recent literature.

(4)    A summative table highlighting the outcomes from previous research is expected at the end of the introduction section

(5)    There is a room to improve the research methodology for publishing in an international journal. Furthermore, the numerical experiments were insufficient.

(6)    The reviewer think some figures related to the computation results should be presented to improve the quality of this paper.

(7)    The conclusion section provides a lack of contributions to this manuscript. Provide the key features, merits, and limitations of the proposed approach to clarify the precise boundary of the algorithms. The implication of the proposed method is also required.

(8)    This paper is generally well written, but I found multiple typographic and editorial errors over the entire manuscript, including the equations. The authors need to proofread again carefully.

Author Response

Please find attached a pdf file with comments to the Editor and to Reviewer 2.
